# *ZmNLR**-7*-Mediated Synergistic Regulation of ROS, Hormonal Signaling, and Defense Gene Networks Drives Maize Immunity to Southern Corn Leaf Blight

**DOI:** 10.3390/cimb47070573

**Published:** 2025-07-21

**Authors:** Bo Su, Xiaolan Yang, Rui Zhang, Shijie Dong, Ying Liu, Hubiao Jiang, Guichun Wu, Ting Ding

**Affiliations:** 1School of Plant Protection, Key Laboratory of Biology and Sustainable Management of Plant Diseases and Pests of Anhui Higher Education Institutes, Anhui Agricultural University, Anhui Province Key Laboratory of Integrated Pest Management on Crops, Hefei 230036, China; subowsm31415@163.com (B.S.); yxiaolan77@163.com (X.Y.); 19505560818@163.com (R.Z.); dongshijie0823@163.com (S.D.); 15855053647@163.com (Y.L.); 12116088@zju.edu.cn (H.J.); 2School of Biological and Food Engineering, Su Zhou University, Suzhou 234000, China; 3National Engineering Laboratory of Crop Stress Resistance Breeding, School of Life Sciences, Anhui Agricultural University, Hefei 230036, China

**Keywords:** *ZmNLR-7*, maize, *Bipolaris maydis*, plant disease susceptibility

## Abstract

The rapid evolution of pathogens and the limited genetic diversity of hosts are two major factors contributing to the plant pathogenic phenomenon known as the loss of disease resistance in maize (*Zea mays* L.). It has emerged as a significant biological stressor threatening the global food supplies and security. Based on previous cross-species homologous gene screening assays conducted in the laboratory, this study identified the maize disease-resistance candidate gene *ZmNLR-7* to investigate the maize immune regulation mechanism against *Bipolaris maydis*. Subcellular localization assays confirmed that the ZmNLR-7 protein is localized in the plasma membrane and nucleus, and phylogenetic analysis revealed that it contains a conserved NB-ARC domain. Analysis of tissue expression patterns revealed that *ZmNLR-7* was expressed in all maize tissues, with the highest expression level (5.11 times) exhibited in the leaves, and that its transcription level peaked at 11.92 times 48 h post *Bipolaris maydis* infection. Upon inoculating the *ZmNLR-7* EMS mutants with *Bipolaris maydis*, the disease index was increased to 33.89 and 43.33, respectively, and the lesion expansion rate was higher than that in the wild type, indicating enhanced susceptibility to southern corn leaf blight. Physiological index measurements revealed a disturbance of ROS metabolism in *ZmNLR-7* EMS mutants, with SOD activity decreased by approximately 30% and 55%, and POD activity decreased by 18% and 22%. Moreover, H_2_O_2_ content decreased, while lipid peroxide MDA accumulation increased. Transcriptomic analysis revealed a significant inhibition of the expression of the key genes *NPR1* and *ACS6* in the SA/ET signaling pathway and a decrease in the expression of disease-related genes *ERF1* and *PR1*. This study established a new paradigm for the study of NLR protein-mediated plant immune mechanisms and provided target genes for molecular breeding of disease resistance in maize. Overall, these findings provide the first evidence that *ZmNLR-7* confers resistance to southern corn leaf blight in maize by synergistically regulating ROS homeostasis, SA/ET signal transduction, and downstream defense gene expression networks.

## 1. Introduction

Plant diseases often cause large-scale reductions in grain production. Breeding disease-resistant varieties is one of the most economical and environmentally friendly means to resist crop diseases. Plants have built a complex and sophisticated immune defense system during the long-term evolution process and precisely regulate their resistance to pathogenic microorganisms through multi-level signaling networks. Its core mechanisms include basic immunity (pattern-triggered immunity, PTI) mediated by cell-surface pattern-recognition receptors (PRRs) and effector-triggered immunity (effector-triggered immunity, ETI) dominated by intracellular nucleotide-binding leucine-rich repeat proteins (nucleotide-binding leucine-rich repeat receptors, NLRs) [1]. PTI activates basic defense responses such as MAPK cascades and callose deposition by recognizing pathogen-associated molecular patterns (PAMPs; such as bacterial flagellar protein flg22 and fungal chitin), while ETI specifically recognizes effector proteins secreted by pathogens through NLR proteins, triggering strong anti-disease phenotypes, such as hypersensitive response (HR) and reactive oxygen species (ROS) bursts [2]. Research breakthroughs in recent years have shown that PTI and ETI have a synergistic mechanism: the two jointly regulate the phosphorylation activation of NADPH oxidase RBOHD, significantly enhancing the ROS synthesis capacity and forming an “immune signal amplifier” effect [3]. For example, in *Arabidopsis*, FLS2 (PTI receptor) and RPS4 (ETI receptor) promote the dual phosphorylation of Ser39/343 sites of RBOHD through interaction, which increases the ROS production by 3–5 times compared with a single pathway, thereby breaking through the antioxidant defense threshold of pathogens [4]. At the level of systemic immune regulation, the three major hormone pathways of salicylic acid (SA), jasmonic acid (JA), and ethylene (ET) constitute a dynamic regulatory network. SA signals regulate the expression of pathogenesis-related genes (*PR* genes) through NPR1 receptor proteins, specifically resisting biotrophic pathogens, while the JA/ET pathway activates the synthesis of defense substances, such as protease inhibitors, through *MYC2/ERF* transcription factors to resist the invasion of necrotrophic pathogens [5]. It is worth noting that there is a fine cross-regulation between hormone pathways: SA can antagonize JA signals by inhibiting the stability of JAZ proteins, while ET can synergistically enhance JA responses by enhancing the activity of *EIN3/EIL1* transcription factors. This “disease resistance signal allocation” mechanism enables plants to optimize defense resource allocation according to the type of pathogen, while maintaining a balance between growth and development and immune response through the SA-JA-ET signal integration node mediated by *EIN2* [6].

As a core member of plant disease-resistance genes (R genes), NLR (nucleotide-binding leucine-rich Repeat) genes play an irreplaceable biological function in effector-triggered immunity (ETI) [7]. Its molecular characteristics are composed of three typical domains: the amino-terminal signal transduction domain (CC or TIR), the central nucleotide binding site (NBS/NB-ARC), and the carboxyl-terminal leucine-rich repeat sequence (LRR). According to the differences in the amino-terminal domain, NLR proteins can be divided into two major categories: CNL (CC-NBS-LRR) and TNL (TIR-NBS-LRR) [8,9]. It is worth noting that dicotyledons have both CNL and TNL types, while monocotyledons (such as wheat) usually only retain the CNL type [10]. Functional studies have shown that the amino-terminal domain (such as CC or TIR) is responsible for downstream immune signal transduction, while the LRR domain specifically recognizes pathogenic effectors through conformational changes [11,12]. Among them, the NBS domain acts as a molecular switch, and its nucleotide binding properties are closely related to the NTPase activity of the STAND (Signal Transduction ATPases with Numerous Domains) superfamily, regulating the protein activation state through the ATP/GTP binding–hydrolysis cycle [13,14]. NLR proteins are not only the core executors of ETI, but their modular structure also provides unique advantages for molecular design. Through strategies such as domain recombination and gene stacking, the precise expansion and persistence of the anti-disease spectrum can be achieved. For example, the wheat stem rust-resistance NLR genes *Sr22*, *Sr35*, *Sr45*, *Sr50*, and *Sr55* were transferred in series into the susceptible variety Fielder, which can give it broad-spectrum resistance to stem rust fungi with different physiological subspecies around the world [15]. Similarly, the wheat *Sr22*, *Sr33*, *Sr35*, and *Sr45* genes were introduced into the barley variety Golden Promise, which successfully obtained specific resistance to stem rust fungus *TTKSK* without affecting agronomic traits [16]. These breakthrough results not only verify the feasibility of NLR genetic engineering but also reveal its great potential in crop disease-resistance breeding [17,18]. The current research focus has shifted to the optimization of multi-gene stacking systems and the analysis of disease-resistance–developmental balance mechanisms, laying a theoretical foundation for the design of the next generation of intelligent disease-resistant varieties.

By examining the disease-resistance phenotype of EMS mutant strains, integrating disease resistance phenotypic transcriptomics, and measuring oxidative stress physiology, this study finished the multi-level functional analysis of the maize disease-resistance candidate gene *ZmNLR-7* (*Zm00001d020117*) in the maize–microbe *Bipolaris maydis* interaction system, which was based on the cross-species homologous gene-screening strategy. This study provided a novel functional module for the molecular design and breeding of maize resistance genes by revealing, for the first time, the molecular mechanism by which *ZmNLR-7* coordinates disease-resistance responses by controlling reactive oxygen homeostasis and the ET/SA signaling pathway.

## 2. Materials and Methods

### 2.1. Bioinformatics Analysis of ZmNLR-7

The full-length coding sequence (CDS) and corresponding amino acid sequence of *ZmNLR-7* (Appendix A) were retrieved from the maize genome database (MaizeGDB; https://www.maizegdb.org, accessed on 18 March 2024). Physicochemical properties of the encoded protein, including molecular weight, isoelectric point (pI), and amino acid composition, were analyzed using the ProtParam tool on the Expasy online platform (https://web.expasy.org/protparam/, accessed on 8 June 2024). Structural domains of ZmNLR-7 protein sequences were predicted via the SMART web server (http://smart.embl-heidelberg.de/, accessed on 10 June 2024). Homologous sequences of ZmNLR-7 in *Zea mays*, *Oryza sativa*, *Sorghum bicolor*, *Setaria italica*, and *Arabidopsis thaliana* were identified through systematic screening of the NCBI GenBank database. Phylogenetic analysis was performed using MEGA7.0 software with the neighbor-joining (NJ) method, and the resultant tree file was imported into the iTOL v6 platform (https://itol.embl.de/, accessed on 16 July 2024) for multidimensional annotation and visualization optimization.

### 2.2. Vector Construction and Subcellular Localization

The *ZmNLR-7* coding sequence was amplified from maize-leaf mRNA using RT-PCR, with primers *pst1*-F (5′-TCCTCTAGAGTCGACCTGCAGATGCCGCATGGTCACGCC-3′) and *pst1*-R (5′-TAAAGCAGGGCATGCCTGCAGTTTTGCACAGCCCTTTTGCA-3′). The verified product was directionally inserted into the *pCAMBIA2300* vector via homologous recombination (Appendix A). Recombinant plasmids were electroporated into Agrobacterium tumefaciens GV3101 (containing pSoup helper plasmid) for agroinfiltration-mediated transient expression. For subcellular localization, Nicotiana benthamiana leaves were infiltrated with recombinant Agrobacterium suspensions and analyzed 48 h post-infiltration, following established protocols [19]. Maize Protoplast Source and Transformation B73 seeds were sown in soil at a depth of ~2 cm and dark-incubated at 28 °C, with soil moisture maintained by regular watering. After 10–11 days of growth, when the second leaf of etiolated seedlings became fully expanded (~12–15 cm in length), protoplasts were isolated and subjected to transformation assays. Successfully constructed plasmids were transiently transformed into maize protoplasts, followed by 12–16 h of dark incubation at 22 °C. Transformed protoplasts were then observed under a confocal laser scanning microscope [20]. Tissue sections were imaged under fluorescence microscopy (488 nm excitation), with *pCAMBIA2300*-GFP empty vector infiltrations serving as controls.

### 2.3. Pathogen Cultivation

To induce sporulation, *Bipolaris maydis* cultures from PDA medium were inoculated onto sterilized sorghum grain medium and incubated for 3–5 days at 26 °C. Surface mycelium was removed, and the colonized grains were evenly spread on trays. UV irradiation (60 min) was applied to trigger conidiation, followed by incubation in a controlled environment (25 °C, 80% RH) for an additional 3–5 days to maximize spore production.

### 2.4. Plant Materials and Treatments

Maize (B73) seeds were germinated in vermiculite (28 °C, 3–5 days) and transferred to pots under controlled conditions (28 °C, 16/8 h light/dark). Prior to inoculation, the conidia of Bipolaris maydis underwent germination rate assessment, with 95.23% germination observed. These activated conidia were subsequently used for infection assays. Five-leaf-stage plants were sprayed with either *Bipolaris maydis* conidial suspension (1 × 10^5^ CFU/mL) or sterile water (both containing 0.1% Tween-20) for experimental and control groups, respectively. Leaf samples were collected at 0, 24, 48, 60, and 72 h post-inoculation (hpi) for RNA extraction and *ZmNLR-7* expression analysis. Three biological replicates (*n* = 3 plants each) were included per group.

Mature maize plants were flash-frozen in liquid nitrogen and stored at −80 °C for RNA extraction and *ZmNLR-7* expression profiling. For pathogen inoculation, field-grown plants at maturity were sprayed with *Bipolaris maydis* conidial suspension (1 × 10^5^ CFU/mL, 0.1% Tween-20), while controls received sterile water with 0.1% Tween-20. Leaf samples were collected at 0–72 hpi for molecular analysis, with disease severity assessed at 7 dpi using standardized disease indices [19,21]. The completely randomized block design included three biological replicates (10 plants each).

### 2.5. RNA Extraction and RT-PCR Analysis

Total RNA was extracted from maize leaf tissues using established protocols [22]. qPCR reactions contained 10 μL of AceQ qPCR SYBR Green Master Mix, 25 ng of cDNA, and gene-specific primers (1 μL each; Appendix A) per reaction. Cycling parameters included initial denaturation (94 °C, 30 s), followed by 40 cycles of 94 °C for 5 s and 55 °C for 15 s. Data normalization and analysis followed the 2^−ΔΔCt^ method [23].

### 2.6. EMS Mutant Identification

The EMS mutants were obtained from the Maize EMS Mutant Library (http://maizeems.qlnu.edu.cn/search/geneid.html, accessed on 6 July 2024). EMS-mutant seedlings were genotyped using primers flanking the target site *ZmNLR-7-1*-F: CATGACACATAACTAGAAGCTG; *ZmNLR-7-1*-R: GTACCCTTCAG-CATTTCCTTC; *ZmNLR-7-2*-F: ATTCCCGGTGAGTAGGTTG; *ZmNLR-7-2-R*: CCAGGCAATCAACGGTAGG, amplifying a 700 bp region (±350 bp from mutation locus). Leaf tissue (1 cm^2^) was collected for genomic DNA isolation via modified *Arabidopsis* protocols [24]. PCR-amplified fragments were sequenced and aligned against wild-type references to identify homozygous lines. Successful editing was confirmed by detecting insertions/deletions (indels) at target loci, with unedited sequences excluded.

### 2.7. EMS Mutant Identification Physiological Parameter Analysis of ZmNLR-7

Maize plants at the five-leaf stage were spray-inoculated with conidia of *Bipolaris maydis* (1 × 10^5^ CFU /mL, 0.1% Tween-20), and the control group was inoculated with an equal amount of Tween-20 supplemented with sterile water. Leaf tissues of the wild type and ZmNLR-7 mutant were collected 48 h after inoculation for oxidative stress analysis. Hydrogen peroxide (H_2_O_2_) accumulation, SOD/POD activity, and malondialdehyde (MDA) content were determined spectrophotometrically, using commercial kits produced by Suzhou Keming Biotechnology Co., Ltd. (Suzhou, China). Data represent the mean ± SD of three biological replicates.

### 2.8. RNA Sequencing (RNA-Seq) Analysis

RNA sequencing was conducted on the leaves of wild-type and ZmNLR-7 mutant maize under untreated and pathogen-challenged (48 hpi with *Bipolaris maydis* conidia) conditions. Libraries were prepared using the Illumina TruSeq™ RNA Library Prep Kit and sequenced by Majorbio (Shanghai, China). Raw reads were processed via the Majorbio Cloud Platform for alignment, differential expression analysis, and functional annotation. Total RNA was extracted from tissue samples, with concentration and purity assessed using a Nanodrop 2000 (Thermo Scientific, Waltham, MA, USA) spectrophotometer, integrity verified by agarose gel electrophoresis, and RNA Quality Number (RQN) determined via Agilent 5300 (Agilent, Santa Clara, CA, USA). Library construction required ≥ 1 μg total RNA per sample at concentrations ≥ 30 ng/μL, RQN > 6.5, and OD_260/280_ ratios of 1.8–2.2. Sequencing depth was 5.88 GB clean data/each sample. Differential expression analysis was performed using DESeq2, with the reference genome *Zea mays* (Assembly: Zm-B73-REFERENCE-NAM-5.0; Source: http://plants.ensembl.org/*Zea_mays*/Info/Index, accessed on 6 March 2025), applying significance thresholds of |log_2_FC| ≥ 1 and adjusted **p**-value (**p**_adj_) <0.05. The primers for quantitative real-time PCR validation are listed in Appendix A.

### 2.9. Statistical Analysis

Statistical significance of results was calculated using one-way ANOVA, followed by Duncan’s test for the least significant difference at a significance level of *p* < 0.05.

## 3. Results

### 3.1. Bioinformatics Analysis of the ZmNLR-7 Gene

In the early stage, the research group used the homologous sequence analysis method to integrate the disease resistance-gene information reported in gramineous crops such as wheat (*Triticum aestivum*) and rice (*Oryza sativa*), and screened the homologous genes of maize (*Zea mays*) using the candidate gene association analysis method. The obtained maize homologous genes were verified, and two candidate genes whose expression was significantly upregulated after induction by *Bipolaris maydis* were screened out and named *ZmNLR-7* (*Zm00001d020117*). *ZmNLR-7* is located on chromosome 7 of maize. The CDS sequence of the *ZmNLR-7* gene is 1578 bp in length, encoding 526 amino acids. The molecular mass of the protein encoded by it is predicted to be 67.3 kDa, and the theoretical isoelectric point is 5.42. Further domain analysis showed that the protein encoded by ZmNLR-7 has a typical NB-ARC domain (Appendix A). The presence of these conserved domains suggests that this gene may be involved in the immune response mediated by nucleotide-binding leucine-rich repeat receptors. ZmNLR-7 belongs to the NLR protein family. The phylogenetic evolutionary tree of this protein family was constructed using MEGA7.0 software. Phylogenetic comparison revealed that the *ZmNLR-7* (*Zm00001d020117*) studied in this experiment had a high homology with maize *ZmNLR-3* (*XM 035960670.1*) and *ZmNLR-4* (*XM 008653508.3*), and it was closely related to rice *OsNLR-8* (*CP101152.1*) (Figure 1).

### 3.2. Analysis of Subcellular Localization of ZmNLR-7

Based on the analysis of the WoLF PSORT online prediction tool, the ZmNLR-7 protein was predicted to be localized in the plasma membrane system. To verify the biological reliability of this prediction, this study conducted a systematic verification through a dual experimental system. First, the p2300-ZmNLR-7 GFP recombinant plasmid and the empty control p2300 GFP (Figure 2A) were injected into Nicotiana benthamiana leaves using the Agrobacterium-mediated transient expression system, and laser confocal microscopy was performed after culturing in the dark at 22 °C for 48–72 h. The results showed that the positive control GFP signal was widely distributed throughout the cell, while the ZmNLR-7 fusion protein showed clear subcellular localization characteristics, with its green fluorescence signal in the nucleus and plasma membrane (Figure 2B). To further confirm the conservation of this localization pattern in the maize cell system, we used the PEG-mediated protoplast transformation system to introduce p2300-ZmNLR-7: GFP and the control vector p1305-GFP into maize protoplasts, and subcellular localization analysis was performed after culturing in the dark at room temperature for 14–16 h. Consistent with the results of transient expression in tobacco, the ZmNLR-7: GFP fusion protein was also localized in the nucleus and plasma membrane in maize protoplasts (Figure 2C).

### 3.3. Analysis of the Expression Pattern of ZmNLR-7

In order to explore the expression of the *ZmNLR-7* gene in different tissues of maize, RT-PCR detection revealed that *ZmNLR-7* was expressed to varying degrees in different tissues of maize, showing significant differences (Figure 3A). *ZmNLR-7* had the highest expression level in leaves, while the expression level was significantly reduced in roots, stems, stamens, female ears, filaments, bracts, and flag leaves. In order to analyze the dynamic regulatory characteristics of *ZmNLR-7* in disease-resistance response, its induced expression pattern was further analyzed by the infection experiment of *Bipolaris maydis*. The temporal expression analysis of 0–72 h after pathogen infection showed that the transcription level of *ZmNLR-7* showed a typical time-dependent response: the expression level reached a peak 48 h after infection, which was significantly upregulated by 11.92 times compared with the untreated state (0 h), and then gradually decreased (Figure 3B).

### 3.4. Obtaining Homozygous Mutants of ZmNLR-7 Maize and Phenotypic Analysis of Response to Infection with Bipolaris Maydis

To verify the genetic stability of *ZmNLR-7*-related EMS mutant lines, this study systematically identified the mutation sites. Using maize B73 wild type and *ZmNLR-7-1* and *ZmNLR-7-2* mutants as materials, leaf tissues were collected at the four-leaf stage to extract genomic DNA. According to the EMS chemical mutagenesis characteristics of the *ZmNLR-7* gene, specific primers were designed in the 350 bp region upstream and downstream of the mutation site (amplification product 700 bp). After PCR amplification and Sanger sequencing analysis (Figure 4A), *ZmNLR-7-1* and *ZmNLR-7-2* had a C→T single base substitution at ChrX:93,688,474 and ChrX:93,694,948, respectively (see Appendix A). *Both ZmNLR-7-1* (EMS4-097d06) and *ZmNLR-7-2* (EMS4-097d0c) lines harbor premature termination codons. Subsequent experiments utilized homozygous EMS mutant lines, and the homozygous mutant lines obtained were used for subsequent studies. Field experiments showed that the *ZmNLR-7* mutant exhibited significant growth and development defects: compared with the wild-type B73, the plant height decreased by 11.21–11.33%, and the thousand-grain weight decreased by about 3.7% (Appendix A). To evaluate the disease resistance in the adult stage, a standardized field design (plant spacing of 40 cm) was carried out on the T3 generation homozygous mutant and the wild type at the experimental base of Anhui Agricultural University. The leaves were inoculated with 1 × 10^5^ CFU/mL conidia suspension of *Bipolaris maydis* (*Bipolaris maydis*) at the tasseling stage (*n* = 15). Seven days after inoculation, the mutant showed a typical disease phenotype: large areas of confluent necrotic spots appeared on the leaves (Figure 4B), and the disease index reached 74.63–84.07, which was significantly higher than that of the wild type (40.74) by 33.89–43.33 (Figure 4C). Further validation through *Bipolaris maydis* inoculation experiments on seedling-stage maize revealed that *ZmNLR-7* mutant lines (*ZmNLR-7-1* and *ZmNLR-7-2*) also exhibited significantly enhanced disease-susceptibility phenotypes (Appendix A), consistent with field observations. The results showed that the loss of *ZmNLR-7* function led to a significant reduction in the resistance of corn to *Bipolaris maydis*.

### 3.5. Determination of Physiological Indicators of ZmNLR-7 Maize Mutants in Response to Infection with Bipolaris Maydis

The relevant physiological indicators of mutants and wild-type plants were determined 48 h after inoculation with *Bipolaris maydis* spore suspension. It was found that the SOD activity of *ZmNLR-7-1* and *ZmNLR-7-2* (891.829 U/g and 577.263 U/g) was 30–55% lower than that of wild-type B73 (1269.42 U/g) (Figure 5A), and the POD activity of mutants (20,690 U/g and 19,640 U/g) was 18–22% lower than that of wild-type B73 (25,190 U/g) (Figure 5B). At the same time, the MDA content in the mutant (97.674 nmol/g and 125.597 nmol/g) was 14–46% higher than that in the wild type (85.936 nmol/g) (Figure 5C), indicating that the membrane lipid peroxidation damage was aggravated. The H_2_O_2_ accumulation of the wild type exceeded that of the mutant, which was 12.51 μmol/g, 8.89 μmol/g, and 10.97 μmol/g, respectively (Figure 5D), revealing that its ROS scavenging ability was impaired. The results showed that the corn *ZmNLR-7* mutant showed significant antioxidant system disorders after infection with *Bipolaris maydis*.

### 3.6. Transcriptomic Analysis of ZmNLR-7 Maize Mutant in Response to Bipolaris Maydis Infection

According to principal component analysis (PCA), different time nodes distinguished samples on the first principal axis (PC1) and explained 62.8% of the differences (Figure 6A), and different treatment groups distinguished samples on the second principal axis (PC2) and explained 9.9% of the differences (Figure 6A), which showed that the collected samples were well representative. In addition, the Venn diagram results showed that the mutants induced by the leaf spot fungus and the wild type had 898 common genes and 3983 mutant-specific genes (Figure 6B). In order to further explore the specific response pattern of the mutants compared with the wild type after infection with the *Bipolaris maydis*, the enrichment analysis of the KEGG pathway was performed. First, for the KEGG results, a total of four KEGG pathways were detected to be significantly enriched 48 h after inoculation (Figure 6C). Specifically, the four signaling pathways of plant hormone signal transduction, the MAPK signaling pathway—plant, phenylpropanoid biosynthesis, and starch and sucrose metabolism—were significantly enriched in the mutant *ZmNLR-7* plants (Figure 6C). It is worth noting that the plant hormone signaling pathway dominates the number of enriched genes, so the focus will be on analyzing the regulatory mechanism of this pathway.

In the mutant strain *ZmNLR-7* infected with *Bipolaris maydis* and the wild-type B73 control group (Bm_ZmNLR-7_VS_B73), ethylene pathway analysis of ethylene synthesis-related genes found that the ethylene synthesis-related gene *MetK* (*Zm00001eb354640*) was downregulated by 0.548 times; the *ACS* gene (*Zm00001eb055950* and *Zm00001eb234010*) was downregulated by 0.258 and 0.684 times, respectively; the *ACO* gene (*Zm00001eb073290* and *Zm00001eb131380*) was downregulated by 0.435 and 0.453 times, respectively; and the ethylene receptor-related genes *EIR* and *ERS* (*Zm00001eb191130*, *Zm00001eb234010*, and *Zm00001eb234010*) were downregulated by 0.548 and 0.684 times, respectively. The *Zm00001eb216810* genes were upregulated by 2.843, 2.255, and 1.841 times, respectively; the downstream transcription factor *EIN2* (*Zm00001eb19690*) was downregulated by 0.412 times; the *EIN3* gene (*Zm00001eb244830* and *Zm00001eb331080*) was downregulated by 0.345 and 0.736 times; the *ERF1* gene (Zm00001eb307550) was downregulated by 0.644 times (Appendix A); and the *CTR1* gene is a negative regulatory element of the ethylene pathway. The upregulation of *CTR1* leads to the inhibition of the activation state of *EIN2*. The downregulation of *EIN2* further blocks the signal transmission to the core transcription factor *EIN3* and the downstream response gene *ERF1*, resulting in the expression levels of response genes such as *EIN3* and *ERF1* also showing a downward trend. For the analysis of the salicylic acid pathway in Bm_ZmNLR-7_VS_B73 (Appendix A), the gene *EDS1* (*Zm00001eb090980*) related to salicylic acid synthesis was downregulated by 0.918-fold; the core receptor *NPR1* (*Zm00001eb399620*) was downregulated by 0.369-fold; and the *TGA* genes (*Zm00001eb147220*, *Zm00001eb197040*, *Zm00001eb236380*, *Zm00001eb280500*, and *Zm00001eb352420*) were downregulated by 0.605, 0.507, 0.522, 0.274, and 0.263 times. The downregulation of the *TGA* genes may lead to the inability to activate the expression of *PR1* and its related genes (*Zm00001eb299340*, *Zm00001eb257300*, *Zm00001eb341580*, and *Zm00001eb341590*). The above results show that under pathogen stress, the synthesis of the salicylic acid signaling pathway and the expression of genes related to the core transduction link of mutant *ZmNLR-7* are inhibited, thus weakening the plant’s ability to respond to disease.

This study further demonstrated the expression changes of key genes in the ethylene signaling pathway and salicylic acid signaling pathway in the *ZmNLR-7* mutant 48 h after inoculation with *Bipolaris maydis* by heat maps. After treatment with *Bipolaris maydis*, the expression levels of ethylene synthesis-related genes, *SAM*, *ACO*, and *ACS,* and other genes in the *ZmNLR-7* mutant line decreased; the ethylene receptor *ETR* negatively regulated ethylene signal transduction, resulting in the upregulation of ethylene receptor-related genes *EIR* and *ERS* genes; and for the *CTR1* gene, as a negative regulatory element of the ethylene pathway, its upregulation led to the inhibition of the activation state of *EIN2*, and the downregulation of *EIN2* further blocked the transmission of signals to the core transcription factor *EIN3* and the downstream response gene *ERF1*, resulting in the expression levels of response genes such as *EIN3* and *ERF1* also showing a downward trend (Figure 6E). At the same time, the expression levels of genes related to salicylic acid synthesis, such as *EDS1*, core receptor *NPR1*, and *TGA*, were all downregulated; downregulation of the *TGA* gene may lead to the inability to activate the expression of *PR1* (Figure 6F). At the same time, the genes related to the jasmonic acid (JA) pathway did not change significantly during the infection process (Appendix A; Appendix A). These transcriptome results indicate that the *ZmNLR-7* mutant may inhibit the key genes of the ethylene and salicylic acid pathways when infected by the leaf spot fungus, resulting in increased susceptibility of corn to the leaf spot fungus.

### 3.7. Fluorescence Quantitative Verification of ZmNLR-7 Transcriptomics

To verify the authenticity of the transcriptome data, this study randomly screened several differentially expressed genes from the transcriptome data for verification, including *Zm00001eb310440* (*2′-deoxymugineic-acid 2′-dioxygenase*), *Zm00001eb137930* (*Protein DOWNY MILDEW RESISTANCE 6*), *Zm00001eb042870* (*UDP-glycosyltransferase 73D1-like*), *Zm00001eb230410* (*terpene synthase 2*), *Zm00001eb350770* (*CBL-interacting protein kinase 14*), *Zm00001eb419870* (*probable protein phosphatase 2C 37*), *Zm00001eb341580* (*pathogenesis-related protein PRMS precursor*), *Zm00001eb244830* (*ETHYLENE INSENSITIVE 3-like 5 protein*), *Zm00001eb122500* (*GA 3-oxidase 1*), and *Zm00001eb401290* (*ethylene-responsive transcription factor RAP2-4*). The experimental results showed that although there were certain differences in expression changes compared with RNA-seq gene expression data, the expression trends of these genes remained highly consistent as a whole (Figure 7), which fully demonstrated the authenticity and reliability of the transcriptome data of this study.

## 4. Discussion

NLR (nucleotide-binding-domain and leucine-rich-repeat-containing) proteins are key receptors in the plant immune system, mainly triggering effector-triggered immune responses (ETIs) by recognizing pathogen effector proteins (effectors) [25]. Plant NLR proteins are the core executors of ETI immunity, and their structural diversity and functional differentiation play a key role in disease resistance. According to the difference in the N-terminal domain, NLR is divided into two categories: CNL and TNL. Monocotyledonous plants (such as corn and wheat) are mainly CNL, while dicotyledonous plants (such as *Arabidopsis*) have both TNL and CNL types [26]. For example, *Arabidopsis* TNL protein RPP1 generates immune signaling molecules (such as 2′cADPR) through the NAD + hydrolysis activity of the TIR domain, activating the EDS1-PAD4-ADR1 complex to drive broad-spectrum resistance [27,28]. The wheat CNL protein TaRPP13L1 relies on the anchor protein TaANK-TPR1 to promote dimerization and enhance resistance to stripe rust [29]. In this study, the maize ZmNLR-7 protein was localized in the nucleus and plasma membrane. The localization of proteins in the cytoplasmic membrane and nucleus is crucial for their functions. The cytoplasmic membrane, as the boundary between the inside and outside of the cell, houses proteins responsible for communication with the external environment, material transport, and intercellular recognition. For example, the wheat protein TaANK-TPR1 targets the NLR protein TaRPP13L1 (which recognizes the pathogen Peronospora Parasitica 13-like 1), promoting its dimerization, thereby triggering intense cell necrosis and ultimately enhancing wheat resistance to stripe rust [29]. In contrast, proteins localized in the nucleus primarily participate in the regulation of gene expression. For instance, the effector protein Pt-1234 from the leaf rust pathogen targets the subdomain C of the wheat transcription factor *TaNAC069*, thereby regulating the role of *TaNAC069* in the defense response against leaf rust in wheat [30]. In this study, through subcellular localization, it was confirmed that ZmNLR-7 can localize in both the cytoplasmic membrane and nucleus. This result suggests that ZmNLR-7 may participate in the plant’s response to small-spot-pathogen stress by recognizing the pathogen and through targeted interactions with the pathogen’s effector proteins. In subsequent studies, this research group will employ experiments such as yeast two-hybrid and BIFC, using maize ZmNLR-7 as bait to screen for effector proteins from the small-spot pathogen that interact with it.

In addition, NLR-type disease-resistance proteins are one of the most important components of the plant immune system, and their activity is precisely regulated by both intramolecular and intermolecular protein interactions (Fine-Tuning Immunity: Players and Regulators for Plant NLRs). For example, the team led by Wang Guanfeng discovered that the maize ZmVPS23-like protein ZmVPS23L interacts with the CC domain of the maize Rp1-D21 (a mutant generated by intramolecular recombination of two CC-NLR-type disease-resistance genes) and relocates it to endosomes, thereby suppressing the hypersensitive response [31]. The wild emmer wheat NLR genes *TdCNL1/TdCNL5* enhance wheat resistance to powdery mildew by coordinately regulating the MLIW170/PM26 locus [32]. In this study, maize ZmNLR-7 participates in plant defense responses through dual localization in the cytoplasmic membrane and nucleus: the plasma membrane regulates ROS metabolism (SOD/POD activity and H_2_O_2_ accumulation), and in the nucleus, it suppresses the core transcription factor *ERF1* of the ethylene signaling pathway, specifically downregulating genes in the SA/ET pathway (such as *NPR1* and *PR1*). This reveals that ZmNLR-7 is involved in plant disease-resistance regulation through hormonal crosstalk. However, it remains unclear which related proteins are directly affected by the absence of ZmNLR-7. Therefore, in subsequent studies, the research group will continue to investigate proteins within maize that can interact with ZmNLR-7, aiming to clarify the modules through which ZmNLR-7 participates in plant disease-resistance responses, thus providing new targets for maize disease-resistance breeding.

Low temperature in *Arabidopsis* inhibits ET signaling by enhancing the expression of SA synthesis genes (*ICS1*/*SID2*), forming an environmental-immune coupling mechanism [33]; while the ethylene response factor *PpERF98* in peach trees inhibits the SA pathway by activating JA/ET signals, increasing sensitivity to pathogens [34]. In this study, upregulation of ET receptor genes (*EIR*, *ERS*) may inhibit the EIN2-EIN3-ERF1 cascade reaction through *CTR1*, weakening SA-dependent resistance. This SA-ET antagonistic pattern is similar to the hormone interaction under low temperature stress [33]. Current research indicates that NLR immune receptors can perform multiple functions. In addition to activating disease-resistance pathways (such as ETI), some members of the NLR family can also participate in plant growth and development processes by regulating hormone signals [35]. For instance, the deletion of the snc1 gene in Arabidopsis leads to the inhibition of plant growth. In this study, the absence of ZmNLR-7 hinders the activation and expression of defense-related genes ERF1 and PR1 in maize, resulting in the suppression of the ethylene (ET) and salicylic acid (SA) disease-resistance signaling pathways, while having a negligible effect on the expression of genes related to jasmonic acid signaling pathways. This is manifested by a significant increase in the sensitivity of the mutant *ZmCIPK-7* to the small-spot pathogen. Plant hormones such as JA and SA can maintain the balance between plant growth and defense through interactions with other plant-growth regulators (PGRs) [35]. Based on this, the research group speculates that the absence of *ZmNLR-7* disrupts the balance between maize growth and defense, enhancing maize susceptibility while inhibiting plant growth (e.g., reduction in plant height and thousand-grain weight). In subsequent studies, this research group will continue to conduct in-depth expression analyses of genes related to plant growth and development in maize *ZmNLR-7* mutants, as well as protein screening for those that interact with ZmNLR-7, with the aim of clarifying the underlying molecular mechanisms of *ZmNLR-7′*s effects on plant growth. We acknowledge the possibility that the reduced plant height and weight observed in the mutants could be linked to NLR pleiotropy. A potential mechanism for these developmental defects may involve NLR-mediated disruption of GA homeostasis or its interaction with other phytohormones, warranting further analysis [36,37].

This study found that the maize *ZmNLR-7* gene specifically regulates the salicylic acid (SA) and ethylene (ET) disease-resistance signaling pathways (independent of the jasmonic acid pathway), revealing a new mechanism by which NLR proteins achieve directional transmission of immune signals through structural plasticity. Based on this feature, gene-editing technologies (such as CRISPR) can be used to target and modify pathogen response elements, and design disease-resistance modules that are only activated when infected by pathogens. Although the existing EMS mutant (*ZmNLR-7*) has verified the function of the gene, chemical mutagenesis may introduce non-targeted mutations. In the future, the role of the gene as a molecular switch in the disease-resistance–yield trade-off will be systematically analyzed through gene knockout/overexpression, providing theoretical and technical support for the breeding of intelligent disease-resistant varieties.

## 5. Conclusions

This study systematically analyzed the molecular mechanism of ZmNLR-7, a member of the maize NLR family, in mediating resistance to *Bipolaris maydis*. This gene constructs a multi-level disease-resistance defense system by maintaining ROS metabolic homeostasis (SOD/POD activity regulation) and integrating the SA/ET signal transduction network. The mutant phenotype confirmed that the loss of *ZmNLR-7* function led to increased membrane lipid peroxidation, inhibition of pathogenesis-related genes, and significant decline in systemic resistance. The results provide a theoretical framework and key targets for analyzing broad-spectrum and durable disease-resistant varieties.

## Figures and Tables

**Figure 1 cimb-47-00573-f001:**
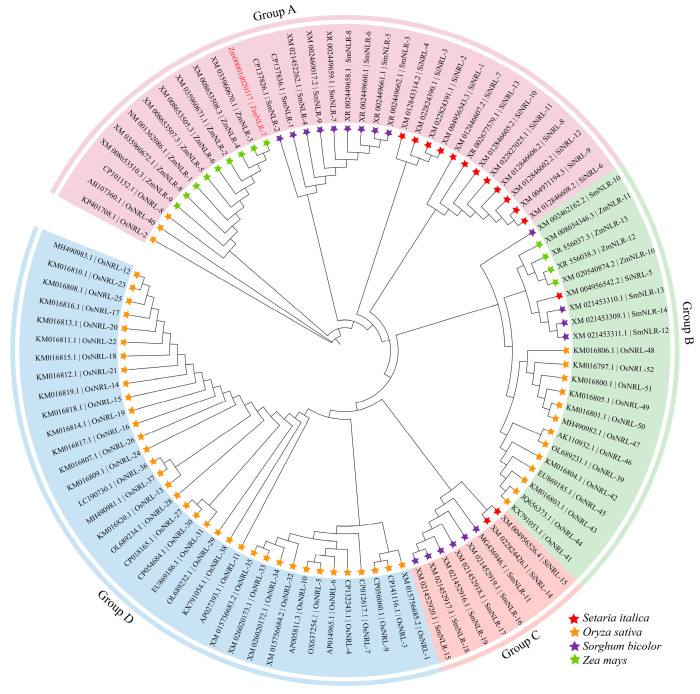
Phylogenetic analysis of ZmNLR-7 from *Triticum aestivum*, *Oryza sativa*, and *Zea may*s. The genes encoding the *ZmNLR-7* were divided into 4 categories. *ZmNLR-7* was marked in red. The species we selected—*Setaria italica*, *Oryza sativa*, *Sorghum bicolor*, and *Zea mays*—are all monocotyledonous plants. Bootstrap values of 1000 replicates are shown as percentages at the branch nodes. Bar = 0.1. The protein-naming method in the figure consisted of autonomously naming with an NCBI number.

**Figure 2 cimb-47-00573-f002:**
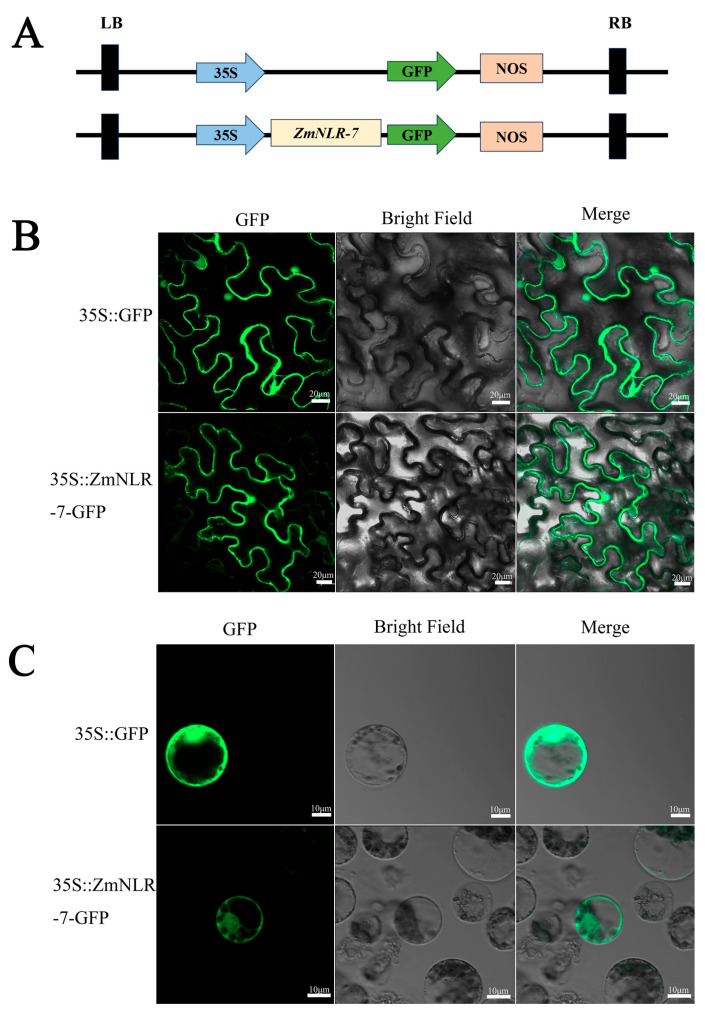
Subcellular localization analysis of ZmNLR-7. (**A**) Schematic representation of the 35S::ZmNLR-7-GFP vector construction for subcellular localization analysis. (**B**) Subcellular localization of ZmNLR-7-GFP in Nicotiana benthamiana leaves; 35S::GFP (empty vector) served as the control. (**C**) Subcellular localization of ZmNLR-7-GFP in maize mesophyll protoplasts. Scale bar: 20 μm, GFP is a green fluorescent protein.

**Figure 3 cimb-47-00573-f003:**
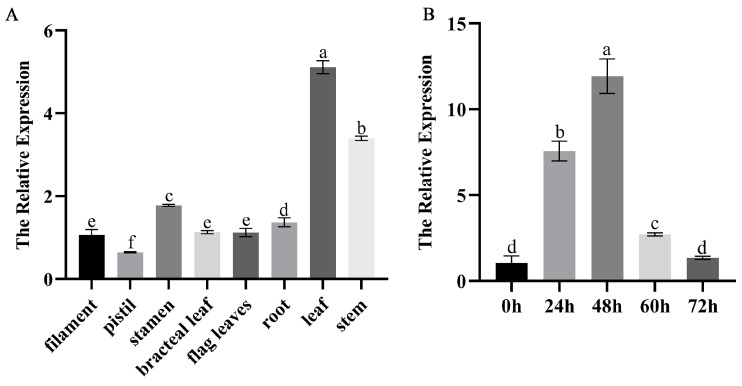
The expression profile of *ZmNLR-7*. (**A**) Tissue-specific expression levels of *ZmNLR-*7 in different maize organs. (**B**) Temporal expression dynamics of *ZmNLR-7* following *Bipolaris maydis* inoculation. Significant differences (*p*-value < 0.05) are indicated with different lowercase letters.

**Figure 4 cimb-47-00573-f004:**
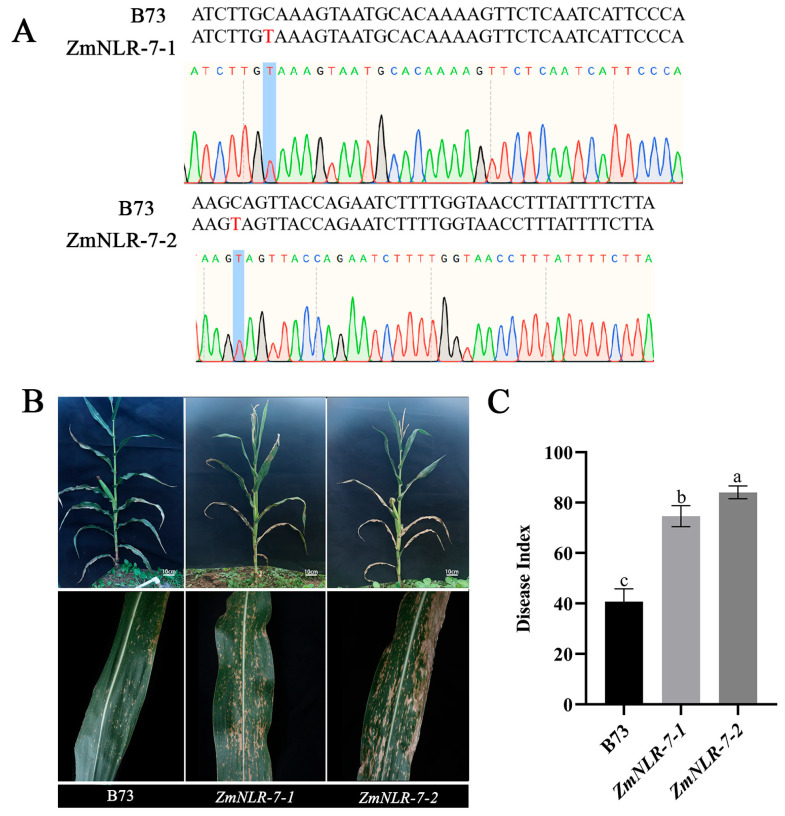
Phenotypic analysis of *ZmNLR-7* mutants in response to *Bipolaris maydis* infection. (**A**) Genotypic identification and sequencing chromatograms of *ZmNLR-7* mutants. (**B**) Disease-resistance phenotypes of wild-type and *ZmNLR-7* mutant lines at 7 days post-inoculation (dpi) with *Bipolaris maydis*. (**C**) Disease index of wild-type and *ZmNLR-7* mutant plants following pathogen challenge. Significant differences (*p*-value < 0.05) are indicated by different lowercase letters.

**Figure 5 cimb-47-00573-f005:**
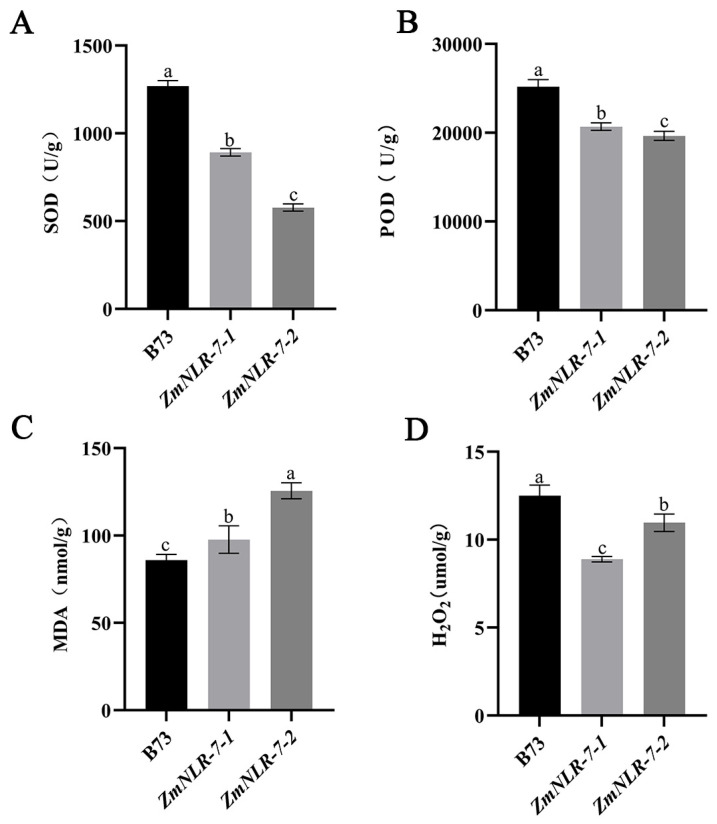
Measurement of disease resistance-related physiological indicators: (**A**) SOD activity; (**B**) POD activity; (**C**) MDA content; and (**D**) H_2_O_2_ content. Significant differences (*p*-value < 0.05) are indicated by different lowercase letters.

**Figure 6 cimb-47-00573-f006:**
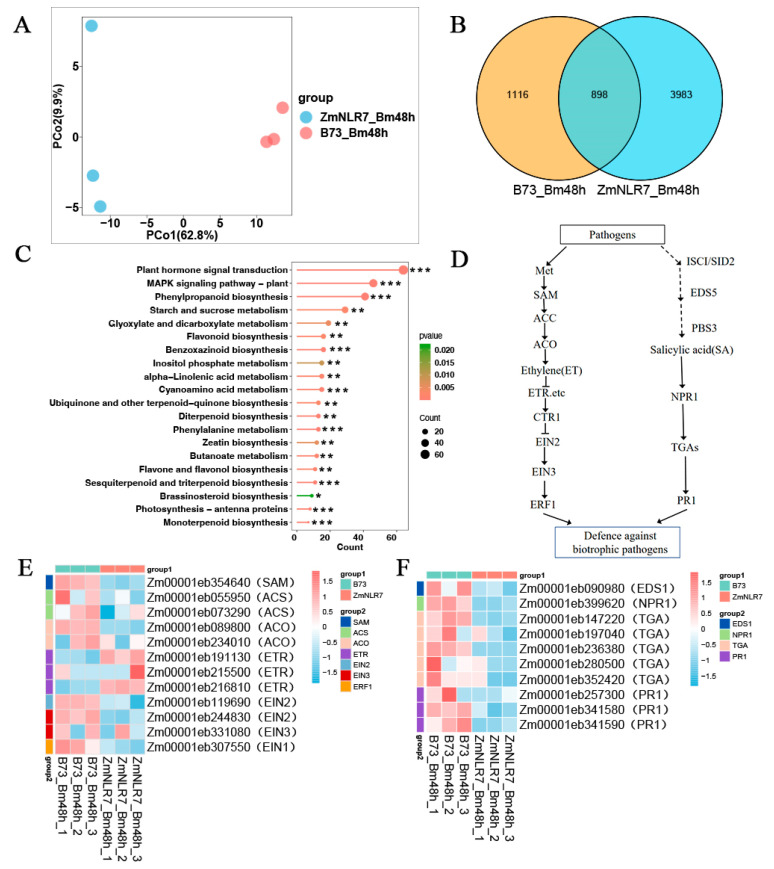
Differential gene expression in maize following *Bipolaris maydis* infection. (**A**) Principal component analysis (PCA) of global transcriptomic profiles across treatments. (**B**) Venn diagram illustrating shared and unique differentially expressed genes (DEGs) among experimental groups (**C**) KEGG pathway enrichment analysis of DEGs in wild type (B73-Bm48 h) versus *ZmNLR-7* mutant (ZmNLR-7-Bm48h) at 48 h post-inoculation with *Bipolaris maydis*. Asterisks indicate significant differences (*: *p* < 0.05; **: *p* < 0.01; ***: *p* < 0.001) (**D**–**F**) Schematic representation of ethylene (ET) and salicylic acid (SA) signaling pathways, with heatmaps showing expression levels of key genes regulated by *ZmNLR-7* in response to pathogen infection.

**Figure 7 cimb-47-00573-f007:**
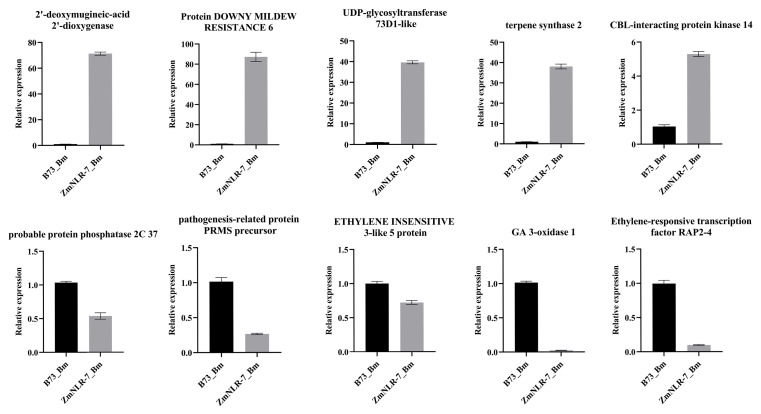
Validation of RNA-Seq data by qRT-PCR.B73-Bm48h and ZmNLR-7-Bm48h denote wild-type and *ZmNLR-7* mutant maize plants, respectively, at 48 h post-inoculation (hpi) with *Bipolaris maydis*. Error bars represent standard deviation (SD) of three independent biological replicates. The left *Y*-axis indicates relative expression levels quantified by quantitative reverse-transcription PCR (qRT-PCR).

## Data Availability

The authors declare that data supporting the findings of this study are available in the article. If raw data files are required, they are available from the corresponding author upon reasonable request.

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
