# Peer review of "ZmNLR-7-Mediated Synergistic Regulation of ROS, Hormonal Signaling, and Defense Gene Networks Drives Maize Immunity to Southern Corn Leaf Blight"

_cimb, 2025, doi:10.3390/cimb47070573_

Round 1

Reviewer 1 Report

Comments and Suggestions for Authors

Dear Su et al.,

Your manuscript, "ZmNLR-7-Mediated ROS and Hormonal Crosstalk Drives Immunity to Southern Corn Leaf," presents a valuable contribution to maize production, particularly in the context of disease resistance.

In order to strengthen the impact of your work, we kindly suggest that the final section of the Discussion or Conclusion highlight the broader significance of this and similar studies. Specifically, we recommend including a reflection on the practical implementation of such findings in crop improvement programs, especially in relation to breeding strategies aimed at enhancing resistance traits.

Furthermore, we encourage you to address the potential implications of this research for global maize production, considering the increasing challenges posed by climate change and emerging pathogens. Emphasizing the translational value of your findings could significantly enhance the relevance of your work to both scientific and agricultural communities.

Sincerely,

Reviewer

Author Response

Response to Referee 1 Comments

Point: Your manuscript, "ZmNLR-7-Mediated ROS and Hormonal Crosstalk Drives Immunity to Southern Corn Leaf," presents a valuable contribution to maize production, particularly in the context of disease resistance.

In order to strengthen the impact of your work, we kindly suggest that the final section of the Discussion or Conclusion highlight the broader significance of this and similar studies. Specifically, we recommend including a reflection on the practical implementation of such findings in crop improvement programs, especially in relation to breeding strategies aimed at enhancing resistance traits.

Furthermore, we encourage you to address the potential implications of this research for global maize production, considering the increasing challenges posed by climate change and emerging pathogens. Emphasizing the translational value of your findings could significantly enhance the relevance of your work to both scientific and agricultural communities.

Response: Thank you for your valuable comments and recognition of our work.We have carefully addressed the reviewers suggestion to highlight the broader significance of our findings. As requested, we added relevant content in the discussion section, mainly involving the significance of the localization of ZmNLR-7 in the cytoplasmic membrane and nucleus for its functionality. This study focuses on the differences in molecular mechanisms between the ZmNLR-7 protein and reported corn NLR family proteins, as well as related research that needs to be conducted by our research group on the ZmNLR-7 protein in the future. (Highlighted in yellow; see line 502-524).

Reviewer 2 Report

Comments and Suggestions for Authors

Overall

This manuscript comprehensively characterizes the maize NLR immune receptor gene ZmNLR-7, demonstrating its critical role in conferring resistance to Bipolaris maydis through coordinated regulation of ROS homeostasis and SA/ET signaling pathways. The multi-level experimental approach (bioinformatics, subcellular localization, EMS mutagenesis, physiological assays, and transcriptomics) robustly validates ZmNLR-7 as a key mediator of disease resistance. The work provides novel mechanistic insights into NLR-mediated immunity in monocots and offers valuable targets for molecular breeding.

Comments

  1. Materials and Methods

(1) In the main text, it’s necessary to supplement details of the mutants, such as how the mutant materials were obtained (whether it was self-generated through EMS mutagenesis or purchased mutant material), the type of mutation (whether the mutations resulted in premature termination), and the ratio of homozygous to heterozygous in the studied materials.

(2) In Section 2.2, it’s recommended to add: (a) maize protoplast source (e.g., 10-day-old seedling leaves); (b) control vector details (e.g., *p1305-GFP*); (c) post-PEG transformation dark incubation (e.g., 14–16 hours).

(3) In Section 2.4, it’s recommended to supplement conidial viability verification (e.g., "≥90% germination rate via hemocytometer").

(4) In Section 2.8, it’s necessary to supplement detail methods of RNA-seq, especially it’s recommended to state the following content: (a) RNA integrity (e.g., RIN ≥8.0); (b) Sequencing depth (e.g., ≥20M reads/sample); (c) Genome version (e.g., B73 RefGen_v5); (d) DEG thresholds (e.g., |log₂FC|>1, FDR<0.05).

  1. Results

(1) In Figure 1, it’s better to annotate monocot/dicot clades (such as triangles), thus highlighting the evolutionary position of ZmNLR-7.

(2) Figure 5 should standardize H₂O₂ units to μmol/g (current: text "U/g" vs. figure "μmol/g").

(3) In Figure 6, it’s better to label gene names (e.g., PR1, ERF1) directly on heatmaps (Figs 6E, 6F), and mark significantly enriched KEGG pathways with asterisks (*p*<0.05) in Fig 6C.

(4) Check Table S5 to include all primers for Figure 7 genes with gene IDs (e.g., ERF1: Zm00001eb307550).

(5) In Figure S2, it’s recommended to add scale bars to field phenotypes (e.g., "Bar = 10 cm").

(6) Figure S3, Table S5 and S6 weren’t referred in the main text, please check the related contents and ensure they should be cited appropriately in the manuscript.

  1. Discussion

(1) In Section 3.4, in terms of reduced plant height/weight in the mutants, it’s recommended to analyze whether it’s linked to NLR pleiotropy, and discuss the potential mechanism of developmental defects in mutants.

(2) It's better to discuss the significance of dual localization: (a) Plasma membrane: potential effector recognition (e.g., TaRPP13L1 anchoring in wheat); (b) Nucleus: direct transcriptional regulation of ERF1 (e.g., NLR nuclear signaling in Arabidopsis).

(3) It’s better to compare ZmNLR-7 with reported ZmNLR-3, thus emphasizing SA/ET pathway uniqueness in ZmNLR-7 and proposing ZmNLR-7 as a modular resistance component in molecular breeding.

Author Response

Response to Referee 2 Comments

Point1This manuscript comprehensively characterizes the maize NLR immune receptor gene ZmNLR-7, demonstrating its critical role in conferring resistance to Bipolaris maydis through coordinated regulation of ROS homeostasis and SA/ET signaling pathways. The multi-level experimental approach (bioinformatics, subcellular localization, EMS mutagenesis, physiological assays, and transcriptomics) robustly validates ZmNLR-7 as a key mediator of disease resistance. The work provides novel mechanistic insights into NLR-mediated immunity in monocots and offers valuable targets for molecular breeding.

1.Materials and Methods

(1) In the main text, it’s necessary to supplement details of the mutants, such as how the mutant materials were obtained (whether it was self-generated through EMS mutagenesis or purchased mutant material), the type of mutation (whether the mutations resulted in premature termination), and the ratio of homozygous to heterozygous in the studied materials.

Response: Thank you for your valuable comments and recognition of our work.The EMS mutants were obtained from the Maize EMS Mutant Library (http://maizeems.qlnu.edu.cn/search/geneid.html) (Highlighted in green; see line185-186).Both ZmNLR-7-1 (EMS4-097d06) and ZmNLR-7-2 (EMS4-097d0c) lines harbor premature termination codons. Subsequent experiments utilized homozygous EMS mutant lines, and the homozygous mutant lines obtained were used for subsequent studies.(Highlighted in green; , see line301-304).

Point2: (2) In Section 2.2, it’s recommended to add: (a) maize protoplast source (e.g., 10-day-old seedling leaves); (b) control vector details (e.g., *p1305-GFP*); (c) post-PEG transformation dark incubation (e.g., 14–16 hours).

Response: Thank you for your valuable advice. We have incorporated the following details into the manuscript as recommended:These additions appear in the Methods section (highlighted in green, see line139-140,144-152) of the revised manuscript.

.

Point3: (3) In Section 2.4, it’s recommended to supplement conidial viability verification (e.g., "≥90% germination rate via hemocytometer").

Response: Thank you for your valuable advice.Prior to inoculation, conidia of Bipolaris maydis underwent germination rate assessment, with 95.23% germination observed. These activated conidia were subsequently used for infection assays.(Highlighted in green; , see line163-166).

Point4: (4) In Section 2.8, it’s necessary to supplement detail methods of RNA-seq, especially it’s recommended to state the following content: (a) RNA integrity (e.g., RIN ≥8.0); (b) Sequencing depth (e.g., ≥20M reads/sample); (c) Genome version (e.g., B73 RefGen_v5); (d) DEG thresholds (e.g., |log₂FC|>1, FDR<0.05).

Response: Thank you for your valuable comments. Thank you for your valuable comments. We have supplemented the RNA-seq methodology details in Section 2.8 These additions have been incorporated into the manuscript (highlighted in green, Lines 210219).

Point5: 2.Results(1) In Figure 1, it’s better to annotate monocot/dicot clades (such as triangles), thus highlighting the evolutionary position of ZmNLR-7.

Response: We are thankful to your positive feedback.This revision is documented in the figure legend (Highlighted in green, see line246-247).

Point6: (2) Figure 5 should standardize H₂O₂ units to μmol/g (current: text "U/g" vs. figure "μmol/g").

Response: Thank you for your valuable comments.The unit "U/g" has been revised to "μmol/g" throughout the manuscript,Corrected in the manuscript (Highlighted in green; see line 338).

Point7: (3) In Figure 6, it’s better to label gene names (e.g., PR1, ERF1) directly on heatmaps (Figs 6E, 6F), and mark significantly enriched KEGG pathways with asterisks (*p*<0.05) in Fig 6C

Response: Thank you for your suggestion, we have revised Figure 6 as follows:

Gene names (e.g., PR1, ERF1) are now directly labeled on the heatmaps (Figs. 6E, 6F).In addition, the significant markers have been marked with asterisks (see Figs).

Point8: (4) Check Table S5 to include all primers for Figure 7 genes with gene IDs (e.g., ERF1: Zm00001eb307550).

Response: Thank you for your suggestion, We have thoroughly checked Supplementary Materials Table S5 (primer list) and confirmed the inclusion of primer sequences for all genes used in Figure 7, along with their standard gene identifiers.

Point9: (5) In Figure S2, it’s recommended to add scale bars to field phenotypes (e.g., "Bar = 10 cm").

Response: Thank you for your suggestion, We have added a scale bar annotation (Bar = 10 cm) to Supplementary Figure S2 (field phenotype images) to indicate the actual dimensions of plant phenotypes. The revised figure has been updated in the manuscript.

Point10: (6) Figure S3, Table S5 and S6 weren’t referred in the main text, please check the related contents and ensure they should be cited appropriately in the manuscript.

Response: Thank you for your correction suggestion, As suggested, Figure S3, Tables S5 and S6 have now been added to the manuscript. Corrections are highlighted in green (see lines 317, 219, 122).

Point11: 3.Discussion(1) In Section 3.4, in terms of reduced plant height/weight in the mutants, it’s recommended to analyze whether it’s linked to NLR pleiotropy, and discuss the potential mechanism of developmental defects in mutants.

Response: Thank you for your suggestion, Current research indicates that NLR immune receptors can perform multiple functions. In addition to activating disease resistance pathways (such as ETI), some members of the NLR family can also participate in plant growth and development processes by regulating hormone signals [1]. For example, the deletion of the snc1 gene in Arabidopsis leads to the inhibition of plant growth. In this study, the absence of ZmNLR-7 obstructs the activation and expression of defense-related genes ERF1 and PR1 in maize, thereby inhibiting the ethylene (ET) and salicylic acid (SA) disease resistance signaling pathways while having a negligible effect on the expression of genes related to the jasmonic acid signaling pathway. This results in a significant increase in the sensitivity of the mutant ZmCIPK-7 to the small spot pathogen. Plant hormones such as JA and SA can maintain the balance between plant growth and defense by interacting with other plant growth regulators (PGRs) [2]. Based on this, the research group speculates that the absence of ZmNLR-7 disrupts the balance between maize growth and defense, enhancing maize susceptibility while inhibiting plant growth (e.g., reduced plant height and thousand-grain weight). According to expert opinions, in subsequent studies, the research group will continue to conduct in-depth expression analysis of genes related to plant growth and development in maize ZmNLR-7 mutants, as well as protein screening of those that interact with ZmNLR-7, aiming to clarify the underlying molecular mechanisms of ZmNLR-7's effects on plant growth. Corrections are highlighted in yellow (see lines 502-524). 

Point12: (2) It's better to discuss the significance of dual localization: (a) Plasma membrane: potential effector recognition (e.g., TaRPP13L1 anchoring in wheat); (b) Nucleus: direct transcriptional regulation of ERF1 (e.g., NLR nuclear signaling in Arabidopsis).

Response: Thank you for your suggestion。我们已在讨论部分补充ZmNLR-7在细胞质膜以及核中定位的意义。We have supplemented the significance of ZmNLR-7 localization in the plasma membrane and nucleus in the discussion section (Highlighted in green; see line 455-473).

Point13:(3) It’s better to compare ZmNLR-7 with reported ZmNLR-3, thus emphasizing SA/ET pathway uniqueness in ZmNLR-7 and proposing ZmNLR-7 as a modular resistance component in molecular breeding.

Response: Thank you for your suggestion.The relevant discussion has been added in the revised manuscript (highlighted in green, Lines 474-494).

  1. Li, X.; Liu, C.; Du, J.; Sun, Y.; Hu, R.; Liu, S.; Xu, Q.; He, X.; Tang, C.-X.; Xu, R.J.C.R. DEAD-box protein SMA1 activates immunity likely through the formation of nuclear condensates with EDS1 in Arabidopsis. 2025, 44.
  2. Verma, K.; Kumari, K.; Rawat, M.; Devi, K.; Joshi, R.J.J.o.S.S.; Nutrition, P. Crosstalk of Jasmonic acid and Salicylic acid with other Phytohormones Alleviates Abiotic and Biotic Stresses in Plants. 2025, 1-23.

Round 2

Reviewer 2 Report

Comments and Suggestions for Authors

I have no other comments and suggestions.